# Chitosan Nanoparticles-Based Ionic Gelation Method: A Promising Candidate for Plant Disease Management

**DOI:** 10.3390/polym14040662

**Published:** 2022-02-09

**Authors:** Nguyen Huy Hoang, Toan Le Thanh, Rungthip Sangpueak, Jongjit Treekoon, Chanon Saengchan, Wannaporn Thepbandit, Narendra Kumar Papathoti, Anyanee Kamkaew, Natthiya Buensanteai

**Affiliations:** 1School of Crop Production Technology, Institute of Agricultural Technology, Suranaree University of Technology, Nakhon Ratchasima 30000, Thailand; huyhoangqct@gmail.com (N.H.H.); fongfangfang_m5430222@hotmail.com (R.S.); c.saengchan5310@gmail.com (C.S.); w.thepbandit@gmail.com (W.T.); narendrakumar.papathoti@gmail.com (N.K.P.); 2Department of Plant Protection, College of Agriculture, Can Tho University, Can Tho 900000, Vietnam; lttoan@ctu.edu.vn; 3School of Chemistry, Institute of Science, Suranaree University of Technology, Nakhon Ratchasima 30000, Thailand; yuiyongyuy@gmail.com (J.T.); anyanee@g.sut.ac.th (A.K.)

**Keywords:** active ingredient, chitosan, ionic gelation method, nanoparticle, plant disease management

## Abstract

By 2050, population growth and climate change will lead to increased demand for food and water. Nanoparticles (NPs), an advanced technology, can be applied to many areas of agriculture, including crop protection and growth enhancement, to build sustainable agricultural production. Ionic gelation method is a synthesis of microparticles or NPs, based on an electrostatic interaction between opposite charge types that contains at least one polymer under mechanical stirring conditions. NPs, which are commonly based on chitosan (CS), have been applied to many agricultural fields, including nanopesticides, nanofertilizers, and nanoherbicides. The CS-NP or CS-NPs-loaded active ingredients (Cu, saponin, harpin, Zn, hexaconazole, salicylic acid (SA), NPK, thiamine, silicon, and silver (Ag)) are effective in controlling plant diseases and enhancing plant growth, depending on the concentration and application method by direct and indirect mechanisms, and have attracted much attention in the last five years. Many crops have been evaluated in in vivo or in greenhouse conditions but only maize (CS-NP-loaded Cu, Zn, SA, and silicon) and soybean (CS-NP-loaded Cu) were tested for manage post flowering stalk rot, Curvularia leaf spot, and bacterial pustule disease in field condition. Since 2019, five of eight studies have been performed in field conditions that have shown interest in CS-NPs synthesized by the ionic gelation method. In this review, we summarized the current state of research and provided a forward-looking view of the use of CS-NPs in plant disease management.

## 1. Introduction

The world population is predicted to reach 9.8 billion by 2050. Demand for food and water will increase, especially in developing countries, where incomes are projected to increase dramatically [1]. Over the period 2010–2050, the total food demand is projected to increase by 35 to 56% and the population at risk of hunger to change from −91 to +8%. Food production is annually affected by climate change as well as pest and disease damage. When affected by climate change, the total food demand and the population at risk of hunger increases, namely, from 30 to 62% and −91 to +30% [2]. Moreover, food production is severely affected by pests and phytopathogens. Plant diseases caused a 16% loss in global crop production between 2001 and 2003 [3]. According to an assessment of [4], phytopathogens cause a 25% yield loss in developing countries. Among them, fungi are the most common (42%), followed by bacteria (27%), viruses (18%), and nematodes (13%). Phytopathogens cause both direct and indirect effects. The consequences of harmful effects are not only loss of productivity; more seriously, but also food quality and toxin production (*Aspergillus*, *Fusarium*) [5]. In severe cases, the potato late blight (*Phytophthora infestans*) break-out caused the historic Irish famine in the mid-19th century [3]. In the research model of [6], when blast rice disease (*Magnaporthe oryzae)* occurred seriously in one country (Thailand) to Southeast Asia, that significantly increased the global rice price, especially for the rice importing countries in sub-Saharan Africa. Disease control or management is essential. Most farmers in developing countries (Laos, Cambodia, and Vietnam) are aware of the risks of pesticides, but they often consider pesticides as an indispensable tool in pest control [7]. A review of pesticide efficacy on controlling arthropod pests, crop pathogens, and weeds indicated that approximately 0.1% of pesticides applied reach their target organism, and pesticides directly affect the health of farmers and pollute the environment [8]. The number of pesticides retained on crops depends on many factors, including the formulation of pesticide used, the volume of spray applied, the type of equipment used, and the quality of the spray [9]. The harmful effect of pesticides on the environment and human health demands research on new harmless means of disease control, to satisfy food security for world’s population. Increasing sustainable agricultural production, reducing food waste, and applying new technology are proposed to ensure food security [10].

In recent years, nanotechnology has emerged as a trend in agricultural production. Nanoparticles (NPs) are very small but powerful and can be applied in many agricultural fields. Nanofertilizers and nanobiotechnology can be used in enhancing crop yields by providing absorbable nutrients and genetic materials, respectively. Water purification, packaging, plant growth, and seed germination were also improved. Furthermore, nanomaterials can be applied to the soil or supplied directly to crops to increase soil health or crop health against adverse environmental factors such as drought, salinity, UV-B, heat, heavy mental, flooding, and biotic stress. Additionally, highly sensitive and precise nanosensors have been used in prediction to make precise farming decisions. Moreover, nanopesticides (nanofungicides, nanobactericides, and nanoinsecticide), nanoelicitors, and nanoherbicides have been applied to protect plants from arthropod pests, phytopathogens, and weeds [11,12,13,14,15,16,17]. Polymers such as gum, mucilage, and chitosan (CS) are naturally derived, readily available, inexpensive, convertible, and biodegrable. Drug-delivery-system-based natural polymers are promising candidates that can be applied in pharmaceutical, biomedical, and agricultural production, with advantages including nontoxicity, water solubility, biocompatibility, and multiple function [18,19,20,21,22]. In the field of plant protection, NPs can be applied as protectors or carriers to load active ingredients to protect plants against insects, fungi, bacteria, and viruses [23]. With their small size, positive charge, and large surface area, NPs have high reactivity and easily penetrate cells through foliage, brand, trunk, root or stick on plant parts [24,25]. According to the review of [26], micronutrient NPs (Cu, ZnO, MgO, and ZnO) and non-micronutrient NPs (Ag) could improve plant growth and inhibit plant pathogens such as *Phoma destructive, P. infestans*, *Rhizopus stolonifer*, *Mucor plumbeus*, *Fusarium oxysporum*, *Botrytis cinerea*, *P. cubensis, P. syringae* pv. *lachrymans,* and *Colletotrichum* spp. In addition, when using NPs as carriers, they can control the released active ingredients to increase uptake, availability of water, and nutrients and reduce the negative environmental effect. Nanopesticides have a high shelf life, site-specific uptake, solubility, low soil leaching, and toxicity [23,27], increasing the efficacy of NPs in controlling plant diseases and enhancing plant growth.

There are two approaches to synthesize NPs, including top-down and bottom-up methods. The physical, chemical, and biological methods can all be used to synthesize NPs and require different equipment and materials, leading to produced NP particles having different characteristics, properties, and functions [28,29]. The ionic gelation technique is a chemical method to synthesize microparticles or NPs based on electrostatic interactions between ions with different charges that was discovered by Calvo et al. (1997) [30,31]. This technique requires polymeric, usually CS and alginate. The cation of CS (R-NH_3_^+^) crosslink with polyanion of sodium triphosphate (TPP) (phosphoric ion) under constant stirring conditions leads to hydrogel formation. The reaction efficiency or properties of the NPs (size, polydispersity indexes determined) differ depending on the ratio of CS and TPP. The process consists of three phases which are solution, aggregation, and opalescent suspension. The materials and instruments for this method can be easily found in conventional laboratories [32,33,34,35,36,37]. Previously, this CS-NPs method was attended in pharmacy to control bacteria on people. The authors of [32] have synthesized CS-NPs-loaded with various metals, including Ag, Cu, Zn, Mn, or Fe, by ionic gelation method. Antimicrobial activity test showed that NPs except Fe could inhibit *Escherichia coli*, *Salmonella choleraesuis*, and *Staphylococcus aureus* at low concentrations, from 3–85 µg/mL, among them, the minimum inhibitory concentration of CS-NPs-loaded Ag and Cu from 3–9 and 9–21 µg/mL, respectively. In addition, the study of [38] showed that CS-NP loaded Ag could inhibit *S. aureus*, *E. coli*, and *Klebsiella pneumonia* with minimum concentrations of 1.69, 1.69, and 3.38 μg Ag/mL, respectively. Moreover, there have been many studies applying NPs synthesized by this method to control plant diseases and enhancing plant growth, most of which are related to CS (shown below). The present article provides a general review of CS-NPs synthesized by the ionic gelation method and their application in plant disease management.

## 2. Ionic Gelation Method

In 1997, two publications by Calvo et al. about the synthesis of NPs by a method called ionic (ionotropic) gelation. There, TPP solution is added to CS and/or diblock copolymer of ethylene oxide and propylene oxide under stirring conditions, leading to the formation of CS-NP particles, with a size 200–1000 nm and zeta potential of 20–60 mV, depending on mass ratio CS/TPP or molecular weight of CS. This hydrogel formation is known due to the electrostatic interactions of the amino group of CS and the polyanion group of TPP. In addition, in these studies, bovine serum albumin (protein), tetanus, and diphtheria toxoid (vaccine) were also successfully loaded into NPs [30,31]. Since then (2021), approximately 11,700 research and review articles related to this method have been published according to the statistics of Google Scholar (Figure 1) [39]. Interestingly, the number of articles has increased year by year, reaching a peak in 2021 with 2090 publications. This shows the researchers’ interest in the ionic gelation method and NPs.

In general, this is a method to form microparticles or NPs which based on electrostatic interaction between opposite charge types that contain at least one polymer under mechanical stirring conditions [37]. On the records of [37,40,41], polymers including CS, carboxymethyl cellulose, collagen, dextran, fibrin, gelatin, gellan gum, hyaluronic acid, sodium alginate, pectin, and anions, including chloride salts (Ba, Ca, Mg, Cu, Zn, and Co), sulfate salts (Na, Mg, or octyl-, lauryl-, hexadecyl-, cetostearyl-), polyphosphate salts (pyro-, tri-, tetra-, octa-, and hexameta-), ferrocyanide, and ferricyanide salts were used for synthesis of NPs. Among them, CS (polymer-cation) and TPP (anion) are most commonly used. Furthermore, drugs or bioactive molecules can be encapsulated into matrix of NPs to increase their efficacy.

CS is a natural polysaccharide, produced by the alkaline deacetylation of chitin, which possesses excellent characteristics, including low toxicity, low cost, biodegradability, biocompatibility, environmental non-toxicity, and adsorption abilities [18,42,43]. With its superiority, CS is used in wastewater treatment, cosmetics, toiletries, food, beverages, agrochemicals, and pharmaceuticals, and its production in the South-East Asian region reaches 1.5 million tons per year [44,45]. The properties of CS can be modified by chemical and/or mechanical processes by hydroxide and/or amide groups, respectively [46]. TPP is also a safe material, commonly used in the synthesis of NPs by the ionic gelation method as a crosslinking agent [37,47].

CS and TPP can be seen as a “legendary” pair of counter ions in the ionic gelation method because of their popularity in studies. Typically, cations and polyanions are released from dissolving CS and TPP in acetic acid and distilled water, respectively. When TPP is dripped into a CS solution, the polyanion (negative charge) bonds to an amino group (positive charge) by electrostatic interaction, which causes CS to undergo a gel ionization process, leading to the formation of NPs that are usually collected by centrifuge [18,48,49]. The primary interactions in ionic crosslinking configuration are H-link and T-link. The H-link is interaction of O^−^ and NH_3_^+^ in the same plane, while the T-link is interaction of nonbrinding oxygen atom and NH_3_^+^ in different plane (Figure 2) [34,37]. The formation of NPs is influenced by CS concentration, CS molecular weight, CS/TPP ratio, drug or bioactive molecules concentration, pH, stirring, and centrifuge (time, speed) [48].

After the synthesis of NPs, their popular features include hydrodynamic diameter, zeta potential, polydispersity index (PDI), morphology, dry state diameter, interaction confirms, encapsulation efficiency (EE), loading capacity (LC), and crystal phase, defined by dynamic light scattering (DLS), scanning electron microscope (SEM), transmission electron microscopy (TEM), Fourier-transform infrared spectroscopy (FTIR), inductively coupled plasma atomic emission spectroscopy (ICP-OES), atomic absorption spectrometry (AAS), ultraviolet visible (UV-Vis), and X-ray diffraction techniques. These features will vary depending on the synthesis condition, see Table 1 and Table 2.

NPs include material with at least one dimension in 1–100 nanometers (nm) range [50]. However, the size of the NPs synthesized by the ionic gelation method is usually greater than 100 nm [51,52,53,54,55,56]. The characteristics of CS-NPs synthesized according to the ionic gelation method are presented in Table 1. In the synthesis, CS-NPs, mass ratios, and volume ratios between CS and TPP vary from 1:10 to 15:1 and 1:10 to 25:1 [32,51,53,57]. CS-NPs smaller than 100 nm are synthesized with CS and TPP mass-to-volume ratios reported in studies of [57] with 2:5 and 1:10 (50 nm), [58] with 5:4 and 1:1 (2.3–7.5 nm), [59] with 6:1 and 3:1 (83.32 nm), [60] with 5:1 and 5:1 (86.8 nm), and [32] with 25:1 and 15:1 (53.99 nm). Similarly, CS-NPs larger than 100 nm were reported in studies of [52] with 3:1 (180.9–595.7 nm), [51] with 1:10 and 1:1 (192.5 nm), [56] with 10:1 and 5:1 (126.2–167.1 nm), 8:1 and 4:1 (493.3–573.1 nm), [54] with 2:5 and 1:1 (100–1000 nm), [53] with 5:1 and 2:1 (238.17 nm), and [55] with 5:1 and 1:2 (204.8–472.1 nm). CS molecular weight also influences NPs formation. As reported by [52], CS-NPs were synthesized from CS low molecular weight (161 kDa) of 180.9 nm in size and were smaller than medium (300 kDa) and high (810 kDa) molecular weights of 309.9 and 339.4 nm, respectively. The authors of [56] had various mass (10:1 and 8:1) and volume (5:1 and 4:1) ratios between CS and TPP with time stirring (60 and 30 min). Results showed that CS-NPs with longer stirring would have smaller sizes in the same mass ratios of 126.2, 167.1, 493.3, and 573.1 nm for 10:1 and 5:1, 8:1, and 4:1 (volume ratio), respectively. This is slightly different from the study of [53] when mass ratio CS:TPP was increased from 5:1 to 20:1 with the same volume ratio, the NPs’ size increased from 238.17 to 1315.37 nm. The authors of [55] conducted research with various sonicate times for 3, 5, 10, and 20 min, and the results showed that the NPs’ size decreased (344.6, 472.1, 261.3, and 204.8 nm, respectively). In addition to centrifuge, the authors of [52] adjusted the pH of the CS:TPP mixture to 4.5–5 to collect NPs. The results showed that the NPs size was greater than that of the centrifugation method for low (180.9 and 225.7 nm) and high (339.4 and 595.7 nm) CS molecular weight and similar to medium molecular weight (309.9 and 301.5 nm), respectively. The PDI of CS-NPs ranged from 0.195 [55] to 1.0 [32]. A low PDI shows a high uniform dispersion of the particles in the solution and vice versa [61]. CS-NPs have PDIs of 0.31–0.52 [52], 0.6 [51], 0.44–0.69 [56], and 0.195–0.57 [55]. Zeta potential is an effective electric charge on NPs’ surface, from −100 mV to 100 mV, representing only NPs’ stability [62]. Zeta potentials of CS-NPs are 21.7–45.6 mV [52], 45.3 mV [51], −28 mV [59], 32.4 mV [63], 20.8–27.8 mV [56], and 51.37 mV [32]. CS-NPs were mostly spherical when observed under SEM or TEM [51,55,57,58,59,60,63], or sphere-like [54,56]. Furthermore, sizes under TEM (dry state) that were smaller than DLS (hydrodynamic size) were shown in the study of [58] with 1.5 nm, [59] with 20–50 nm, [63] with 10–30 nm, and [54] with 100 nm, while DLS was 2.3–7.5 nm, 83.32 nm, 89.8 nm, and 100–1000 nm, respectively. In CS-NPs, the interaction between ammonium group of CS and polyphosphoric of TPP was determined by FTIR with peaks (cm^−1^) of 3428, 1580 [57], 3288, 1647 [58], 1636, 3410 [51], 1648.84, 1527.35 [59], 1563 [60], and 3421.2 [53]. The crystal phase of CS-NPs is amorphous and has been identified by X-ray diffraction [58,59]. UV-Vis is not common in determining properties of CS-NPs synthesized according to the ionic gelation method. As reported by the authors of [59] and [54], CS-NPs absorb at wavelengths of 295 and 320 nm, respectively. Interestingly, the authors of [63] synthesized CS-NPs by CS and anionic protein of *Penicillium oxalicum* by mass and volume ratio by 7.5:0.108 and 5:2 under stirring for 30 min. Hydrodynamic size, dry state size, PDI, and zeta potential were 89.8 nm, 10–30 nm (spherical), 0.225, and −37 mV, respectively, and the peaks of 1602.8, 1564.18, and 1403.5 cm^−1^ characterized the binding of proteins and CS. These NPs absorb at wavelength of 285 nm and are amorphous.

To improve application efficiency, CS-NPs can load drugs and bioactive molecules (active ingredients), depending on the purpose. These NPs were synthesized by adding drugs and bioactive molecules solution to CS and TPP during the gelation process (incorporation) or after that (incubation) [18,42,49,50]. This is shown in Table 2. The active ingredient can be metal ion (Cu, Ag, Zn, Mn, and Fe) [32,51], drug (gentamicin-salicylic acid (SA) complex and Ag-Furosemide complex) [55,64], protein (Harpin from *Pseudomonas syringae* pv. *syringae*), agrochemical (Hexaconazole) [58], hormone (SA) [65], or other bio-molecules (saponin, thiamine, or *Achillea millefolium* extract) [51,66,67]. The mass ratio between the active ingredient and the CS or TPP can be smaller or larger, which affects the characters of the NPs. Metal ions (Ag, Cu, Zn, Mn, and Fe) are added to CS: TPP (1:10 or 15:2) at a rate such that the final ion concentration reaches 0.012% [32,51]. Ag-Furosemide complex was added to CS:TPP, with ratios of 0.005:10:2 and 0.01:10:2. The DLS of the two NPs was 210.5 nm, PDI 0.232, and 41.5 mV; and 197.1 nm, PDI 0.234, and 36.7 mV, respectively [55]. When changing the mass ratio TPP:CS by 1:3, 1:4, 1:5, 1:6, and 1:7 (keeping the gentamicin-salicylic complex ratio), the DLS of NPs changed with decreasing size and increasing zeta potential (343.3 nm, PDI 0.41, 34.26 mV; 217.7 nm, PDI 0.275, 35.77 mV; 180.0 nm, PDI 0.235, 37.12 mV; 172.2 nm, PDI 0.308, 39.44 mV; and 150.8 nm, 0.237 PDI, 42.43 mV, respectively) [64]. On the other hand, the results of various mass ratios between TPP and CS:Hexaconazole by 1:5:10, 2:5:10, 4:5:10, and 8:5:10 showed that the sizes of the NPs decreased with respect to their ratios (220.2, 164.2, 68.1, and 6.5–18.1 nm, respectively) [58]. The Harpin protein (*P. syringae* pv. *syringae*) was also loaded into CS-NPs, with size of 133.7 nm and zeta potential at 48.6 mV, with an initial mass ratio of 1: 100: 20 [60]. When adding SA to CS: TPP at the ratio 1: 4: 2, the DLS was 368.7 nm, PDI 0.1, and +34.1 mV [65]. The authors of [51] and [66] added Saponin and Thiamine to CS:TPP at the ratio 1:2:20 and 25:24:4, and DLS of the two NPs was 373.9 nm (2 peaks), PDI 1.0, +31 mV and 596 nm, 37.7 mV, respectively. In a study by [67], CS-NP was loaded with *A. millefolium* extract by mix the extract (semi-solid form) with 0.1% CS solution to obtain 20% before adding 1% TPP solution. This led to the formation of NPs with a size of 118 nm but containing 3 peaks (10, 122, and 712 nm). CS-NPs-loaded mental ions have a compact polyhedron shape, while CS-NPs-loaded saponin, SA, and gentamicin-SA complex were spherical when observed under SEM and TEM [51,64,65]. The size of CS-NPs-loaded active ingredient when recorded under TEM is sometimes larger than the size specified by DSL. CS-NPs-loaded hexaconazole with initial CS:TPP:hexaconazole ratios of 5:1:10 and 5:2:10 had dried state sizes at 271.4 nm and 168.5 nm, while DLS is 220.2 and 164.2 nm, respectively. However, at the ratio of 5:4:10 and 5: 8:10, the TEM sizes were 32.3 and 8.1 nm, compared with DLS with 68.1 and 6.5–18.1 nm (2 peaks), respectively [58]. In contrast, CS-NPs-loaded thiamine was 596 nm by DLS but 10–60 nm by TEM [66]. Not only using TPP as anions, the authors of [68] also synthesized a novel conductive bio-composite membrane by combining CS and phosphotungstate anions on an aluminum substrate using the the ionic gelation method. Interaction in these NPs can also be determined by FTIR. CS-NPs-loaded Cu is characterized by peaks of 1631 and 1536 cm^−1^ for amide (-CONH_2_) and primary amide (-NH_2_), respectively [51]. Peaks of 1345 and 1095 cm^−1^ feature Harpin protein assigned to C-N and C-O stretch [60]. Peaks of 3218 and 3430 cm^−1^ characterize a hydrogen bonding between three chemicals in CS-NPs-loaded hexaconazole [58]. CS-NPs-loaded saponin was characterized by peak 3430 cm^−1^ for the hydrogen bonding between saponin and CS, and 1536 cm^−1^ for amide linkage between saponin and CS-NPs [51]. The peak at 1317 cm^−1^ featured an interaction between -COOH of SA and primary amide (-NH_2_) of CS in a CS-NPs-loaded SA [65]. In CS-NPs-loaded gentamicin-SA complex, a peak at 3423 cm^−1^ characterizes the hydrogen bonding between -OH group bending of gentamicin and CS, two peaks of 1542 and 1637 cm^−1^ for interaction between NH_3_^+^ of CS and TPP, and 1300 cm^−1^ for CN bending between COOH of SA and primary amide of CS [64]. Additionally, the peak of 1657 cm^−1^ characterizes the binding of thiamine and CS in these NPs [66]. The crystal phase of the NPs can also be identified by the X-ray diffraction technique. CS-NPs-loaded Ag-Furosemide complex was amorphous, while the crystalline peak of hexaconazole was clearly embedded in the amorphous phase of CS [58]. In addition, peak 2θ of 10°–20° and 18°–30° was recognized for SA and CS, respectively [65]. UV-Vis is seldom used, and only CS-NPs-loaded with 267 nm absorption thiamine was reported by [66]. In general, the main steps for synthesizing and building the CS-NPs-loaded active ingredients using the ionic gelation method are shown in Figure 3. Parameters can be optimized to suit each laboratory’s conditions.

The ionic gelation method requires simple, easy-to-find, and expensive materials and equipment, so it can be done easily, mildly, and quickly in normal laboratories. In addition, the mechanism based on electrostatic interaction instead of chemical reaction leads to no need to use organic solvents, thus avoiding potential toxicity of chemicals or reagents. However, the disadvantage of this method is that it is not easy to produce uniformly sized NPs, and research on other polymers (not CS) is limited [18,37,40,49].

**Table 1 polymers-14-00662-t001:** Character of CS-NPs synthesized by ionic gelation method.

Mass Ratio	Volume Ratio	Condition Synthesis	DLS	SEM, TEM	FTIR (cm^−1^)	UV (nm)	XRD	Reference
2:5	1:10	CS 0.2%: TPP 0.05% (1:10), 25 °C, pH 4	50 nm	Spherical	3428, 1580: hydrogen bonding between polyphosphoric of TPP and ammonium group of CS	-	-	[57]
3:1	3:1	3 mL of 0.5% CS varying between (A) low molecular weight (Mw = 161 kDa), (B) medium molecular weight (Mw = 300 kDa), (C) high molecular weight (Mw = 810 kDa), and 1 mL of 0.5% TPP, centrifuge 25,000 rpm for 30 min	(A) 180.9 nm, PDI 0.31, 45.6 mV (B) 309.9 nm, PDI 0.46, 33.2 mV (C) 339.4 nm, PDI 0.52, 21.7 mV	-	-	-	-	[52]
3:1	3:1	3 mL of 0.5% CS (pH 4.7–5) varying between (D) low molecular weight (Mw = 161 kDa), (E) medium molecular weight (Mw = 300 kDa), (F) high molecular weight (Mw = 810 kDa), and 1 mL of 0.5% TPP, adjust pH to 4.5–5, and discard supernatant to collect CS-NPs	(D) 225.7 nm, PDI 0.44, 33.4 mV (E) 301.5 nm, PDI 0.2, 20.2 mV (F) 595.7 nm, PDI 0.92, 16 mV	-	-	-	-	
5:4	1:1	0.25 g CS and 0.2 g TPP/40 mL, pH 3.6, 2% TWEEN 80, and 40,000 rpm for 10 min	Bimodal particle with 2.3 and 7.5 nm	1.5 nm Spherical	3288, 1647: hydrogen bonding between amino of CS and phosphate of TPP		Amorphous	[58]
1:10	1:1	0.1% CS and 1% TPP with ratio 1:1; 10,000 rpm for 10 min and ultrasonication	192.5 nm PDI 0.6 +45.33 mv	Spherical	1636, 3410: hydrogen bonding	-	-	[51]
6:1	3:1	0.5% CS (pH 5) and 0.25% TPP with ratio 3:1, 10,000 rpm for 10 min	83.32 nm PDI 0.31 −28 mV	Spherical 20–50 nm	1648.84, 1527.35: interaction between ammonium group of CS and polyphosphoric group of TPP	295 nm	Amorphous	[59]
5:1	5:1	5 mL of 0.1% CS (pH 5.5) and 1 mL of 0.1% TPP, 20,000 rpm for 30 min	86.8 nm +32.4 mV	Spherical	1563: interaction of amide and phosphate	-	-	[60]
69.4:1	5:2	15 mL of 0.5% CS (pH 4.8) and 6 mL of 0.018% anionic protein of *P. oxalicum*, stirring 30 min, and centrifuge 10,000× *g* for 10 min	89.8 nm PDI 0.225 −37 mV	Spherical 10–30 nm	1602.8, 1564.18, 1403.5: binding of Protein and CS	285 nm	Amorphous	[63]
10:1 8:1	5:1 4:1	0.2% Hydrolyzed CS (by chitinase from *Burkholderia cepacia* E76) and 0.1% TPP with varying ratio (A) 5:1 and stirring 60 min, (B) 5:1 and stirring 30 min, (C) 4:1 and stirring 60 min, and (D) 4:1 and stirring 30 min	(A) 126.2 nm, PDI 0.44, 27.8 mV (B) 167.1 nm, PDI 0.47, 25.4 mV (C) 493.3 nm, PDI 0.69, 22.9 mV (D) 573.1, PDI 0.54, 20.8 mV	Spherical-like	-	-	-	[56]
2:5	1:1	0.1% CS, 0.25% TPP with ratio 1:1, centrifuge 10,000 rpm for 10 min, and ultrasonication 28% pulse for 100 s at 4 °C	100–1000 nm	100 nm Spherical-like High porous surface	1576: NH_2_ bond (wagging) 1412: C-H bending vibration of alkyl group	320 nm	-	[54]
5:1 10:1 15:1 20:1	2:1	10 mL of 0.1% TPP, 20 mL of CS varying between (A) 0.25%, (B) 0.5%, (C) 0.75% and (D) 1%; stirring 8 h, and sonication 45 min	(A) 238.17 nm (B) 575.2 nm (C) 706.01 nm (D) 1315.37 nm	-	3421.2: interaction between phosphate and NH_2_	-	-	[53]
15:2	25:1	25 mL of 0.3% CS, 1 mL of 1% TPP; stirring 20 min, sonication 1.5 kW for 30 min, and centrifuge 12,000× *g* for 10 min	53.99 nm PDI 1.0 51.37 mV	-	-	-	-	[32]
5:1	1:2	10 mL of 1% CS, 20 mL of 0.1% TPP, pH 5.5, stirring 1000 rpm for 5 min, sonication 30% amplitude, varying between (A) 3, (B) 5, (C) 10, and (D) 20 min	(A) 344.6 nm, PDI 0.57, 44.1 mV (B) 472.1 nm, PDI 0.507, 42.9 mV (C) 261.3 nm, 0.195 PDI, 42.8 mV (D) 204.8 nm, 0.205 PDI, 36.0 mV	Globular	-	-	-	[55]

Note: CS: chitosan; DLS: dynamic light scattering; FTIR: Fourier-transform infrared spectroscopy; NPs: nanoparticles; PDI: polydispersity index; SEM: scanning electron microscope; TEM: transmission electron microscopy; TPP: sodium tripolyphosphate; UV: ultraviolet visible; XRD: X-ray diffraction.

**Table 2 polymers-14-00662-t002:** Character of CS-NPs-loaded active ingredients synthesized by ionic gelation method.

NPs	Mass Ratio	Volume Ratio	Condition Synthesis	DLS	SEM, TEM	FTIR (cm^−1^)	UV (nm)	XRD	Reference
CS-NP-loaded copper (Cu)	1:10	1:1	0.1% CS and 1% TPP with ratio 1:1, added 0.01% CuSO_4_ to final concentration of Cu^2+^ 0.012% in mixture, 10,000 rpm for 10 min, and ultrasonication	196.4 nm PDI 0.5 +88 mv	Compact polyhedron	1631: -CONH_2_ 1536: -NH_2_	-	-	[51]
CS-NP-loaded mental ion (Ag, Cu, Zn, Mn, and Fe)	15:2	25:1	25 mL of 0.3% CS, 1 mL of 1% TPP, and salt solution at 0.3% added to mixture to ion final concentration 0.012%; stirring 20 min, sonication 1.5 kW for 30 min, and centrifuge 12,000× *g* for 10 min	(Ag) 90.29 nm, 92.05 mV (Cu) 121.9 nm, 88.69 mV (Zn) 210.9 nm, 86.65 mV (Mn) 102.3 nm, 75.74 mV (Fe) 95.81 nm, 71.42 mV	-	-	-	-	[32]
CS-NP-loaded Silver-Furosemide complex	10:2:0.01	1:2	10 mL of 1% CS, 20 mL of 0.1% TPP, pH 5.5, stirring 1000 rpm for 5 min, and sonication 30% amplitude for 10 min. (A) 5, (B) 10 mg Silver-Furosemide complex was mixed with TPP solution	(A) 210.5 nm, PDI 0.232, and 41.5 mV (B) 197.1 nm, PDI 0.234, and 36.7 mV	-	-	-	Amorphous	[55]
CS-NP-loaded Harpin (*P. syringae* pv. *syringae*)	100:20:1	10:2:1	5 mL of 0.1% CS (pH 5.5) and 1 mL of 0.1% TPP, 0.5 mL of 0.01% Harpin, and 20,000 rpm for 30 min	133.7 nm +48.6 mV	-	1345, 1095: Harpin assigned to C-N stretch and C-O stretch in CS-NP-loaded Harpin	-	-	[60]
CS-NP-loaded Hexaconazole	5:1:10 5:2:10 5:4:10 5:8:10	5:2:5	100 mL of 0.5% CS and 100 mL of 1% hexaconazole, 2% TWEEN 80, 40 mL of TPP varying between (A) 0.25%, (B) 0.5%, (C) 1%, and (D) 2%	(A) 220.2 nm (B) 164.2 nm (C) 68.1 nm (D) 2 peaks (6.5 and 18.1 nm)	(A) 271.4 nm (B) 168.5 nm (C) 32.3 nm (D) 8.1 nm	3218: hydrogen bonding of 3 chemicals	-	Crystalline peak of hexaconazole clear embedded in amorphous phase of CS	[58]
CS-NP-loaded saponin	2:20:1	10:10:1	0.1% CS, 1% TPP and 0.5% saponin with ratio 10:10:1, 10,000 rpm for 10 min, and ultrasonication	373.9 nm (2 peaks) PDI 1.0 +31 mV	Spherical	1560: amide linkage between saponin and CS-NPs 3430: hydrogen bonding between saponin and CS	-	-	[51]
CS-NP-loaded SA	4:2:1	1:1:1	0.4% CS, 0.2% TPP and 0.1% SA with ratio 1:1:1	368.7 nm PDI 0.1 +34.1 mV	Spherical and porous	1541, 1639: acetoxy group of SA 1317: interaction between COOH group of SA with primary amide of CS		Peak at 2θ of 10°–20° denoted SA Peak 2θ of 18°–30° confides CS	[65]
CS-NP-loaded gentamicin (GM) and SA	-	-	0.1% SA and 0.2% GM with ratio 3:2, 0.2% CS (pH 5). A mass TPP solution added into CS with ratio varying between (A) 1:3, (B) 1:4, (C) 1:5, (D) 1:6, (E) 1:7; stirring 1 h, and centrifuge 16,000 rpm for 30 min	(A) 343.3 nm, PDI 0.41, 34.26 mV (B) 217.7 nm, PDI 0.275, 35.77 mV (C) 180.0 nm, PDI 0.235, 37.12 mV (D) 172.2 nm, PDI 0.308, 39.44 mV (E) 150.8 nm, PDI 0.237, 42.43 mV	-	-	-	-	[64]
	-	-	CS/TPP ratio 4:1, pH 5.0, drug-to-polymer ratio 1:4, and feed ratio of SA to GM 1.5:1.0.	180 nm PDI 0.235 37.12 mV	Spherical 200 nm	3423: hydrogen bonding between -OH group bending of GM and CS 1542, 1637: interaction between NH_3_^+^ of CS and TPP 1300: C-N bending (interaction between -COOH of SA and primary amide of CS)	-	-	
CS-NP-loaded Thiamine	24:4:25	24:8:25	375 mg Thiamine/75 mL, 360 mg CS/72 mL and 60 mg TPP/24 mL, stirring overnight, and centrifuge 10,000× *g* for 30 min	596 nm 37.7 mV	10–60 nm	1657: binding of Thiamine to CS	267 nm	-	[66]
CS-NP-loaded *A. millefolium* extract	1:5:200	2:1:-	*A. millefolium* extract (semi solid form) added into 10 mL of 0.1% CS to obtain final concentration at 20%, 5 mL of 1% TPP, stirring 2 h and centrifuge 10,000× *g* for 10 min	118 nm with 3 peaks (10, 122 and 712 nm)	Spherical 4.15–100 nm	3281.73, 2163.36 and 1636.78: interaction in NP	417 nm	-	[67]
CS-NP-loaded SA	2:1:2	1:1:1	0.4% CS, 0.2% TPP, and 0.2% SA with ratio 1:1:1	89.86 nm PDI 0.36 22.27 mV	Spherical	3421: NH_2_ stretch 1640: CO-NH_2_ 1540: NH_2_ bend 1314: COOH and NH_2_ 895: Anhydro glycoside	-	-	[69]
CS-NP-loaded silver (Ag)	5:5:0.51	1:1:1	0.5% CS, 0.5% TPP, and 3 mM silver nitrate with ratio 1:1:1	249 nm PDI 0.53 13.53 mV	Spherical	3423: NH_2_ stretch 1643: CO-NH_2_ 1542: NH_2_ bend 894: Anhydro gly-coside	-	-	[69]

Note: CS: chitosan; DLS: dynamic light scattering; FTIR: Fourier-transform infrared spectroscopy; NPs: nanoparticles; PDI: polydispersity index; SEM: scanning electron microscope; TEM: transmission electron microscopy; TPP: sodium tripolyphosphate; UV: ultraviolet visible; XRD: X-ray diffraction.

## 3. Application of CS-NPs-Based Ionic Gelation Method in Plant Disease Management

With its advantages, CS-NPs synthesized according to the ionic gelation method has been applied in many fields, including pharmaceuticals, new materials, and agriculture (nanopesticides, nanofertilizers, and nanoherbicides) [18,42,49,55,64,68].

For the management of plant diseases, CS-NPs can be applied as protectants (nano pesticides) and carriers (fungicides, insecticides, herbicides, plant hormones, elicitors, and nucleic acids) [18,23,70]. In particular, using CS-NPs as a delivery system is of special interest because it can load and protect the ingredients surrounding the environment and release them to the target site uptake of the plants [18]. In addition, with the basic properties of NPs having a small size and high contact area, CS-NPs or CS-NPs-loaded active ingredients can be easily penetrated and permeated into the membrane of phytopathogens or enhanced plant tissues uptake, resulting in an increased control or defense response activity, respectively [49]. Therefore, these NPs can be used directly and indirectly to manage plant diseases.

Studies using CS-NPs synthesis by ionic gelation method in plant disease management are shown in Table 3 and Figure 4.

### 3.1. Chitosan Nanoparticles (CS-NPs)

#### 3.1.1. Directly

Under *in vitro* conditions, the authors of [52] determined that the minimum inhibitory concentration of CS-NPs prepared by centrifuge and pH change method at 0.05 and 0.09% could inhibit growth of *F. graminearum* at 31.97% and 29.67%, respectively. Furthermore, the authors of [52] also showed that CS-NPs were originated from CS low molecular weight, which has a higher inhibitory effect than CS height molecular weight at the same concentration. CS-NPs may or may not inhibit pathogens. CS-NPs may inhibit mycelial growth of *Pyricularia grisea* (65%) at 0.1% [71], *Colletotrichum gloeosporioides* (85.7%) [56], *C. gloeosporioides* (37.8%), *Phytophthora capsica* (50.7%), *Sclerotinia sclerotiorum* (39.5%), *Fusarium oxysporum* (50.3%), *Gibberella fujikuori* (56.3%) at 0.5% [54], *P. grisea* (92%), *Alternaria solani* (87%), *F. oxysporum* (72%) with amount of 100 µg [63], *A. alternata* (80.1–82.2%), *R. solani* (32.2–34.4%) at 0.06–0.1%, *M. phaseolina* (84%) at 0.1% [51], *A. solani* (10, 70%) at 0.03, 0.04% [72], *Alternaria tenuis* (67.67%), *Aspergillus niger* (62.75%), *Aspergillus terreus* (74.67%), *Baeuvaria bassiana* (76.08%), *F. graminearum* (60.37%), *F. oxysporum* (66.60%), *Sclerotium rolfsii* (37.41%) at 800 ppm, and the zearalenone produced by *F. graminearum* [73]. Furthermore, CS-NPs were 0.014% (in acetate buffer), the lysis zone diameter of *Clavibacter michiganensis* and *Fusarium graminearum* were 29.5 and 20.0 mm, and CS was 22.5 and 18.0 mm, respectively [72]. Moreover, 0.2 mL of CS-NP at 125 ppm could inhibit mycelium of *F. graminearum* by 44.3%, higher than fungicide (8-hydroxy quinoline) at 42.33% [73]. In addition, CS-NPs also inhibited spore germination of *C. gloeosporioides* (61.2%) [56] and *A. alternata* (84.4–87.1%) at 0.06–0.1% [51]. On the other hand, CS-NPs do not inhibit mycelial growth, spore germination, and sporulation of *P. grisea,* even at a concentration of 0.1% [59,71]. A study by [54] showed that OD_600 nm_ of *Erwinia carotovora* subsp. *carotovora* strains 113114, 113154, and YKB133061, and *Xanthomonas campestris* pv. *vesicatoria* strain 11,154, were reduced by 41.3, 55.5, 48.5, and 52.1% when treated with CS-NPs at 0.5%; interestingly, they were also reduced by 64.7, 76.3, 78.0, and 73.8% when CS-NPs were treated at 0.05%, respectively. Furthermore, CS-NPs at 2 mL/L inhibited anthracnose disease at 87.5 and 75% for chili and at 50 and 10% for papaya, by using the preventative and curative treatments under *in vivo* conditions, respectively [56].

#### 3.1.2. Indirectly

Pre-treatment of CS-NPs at 0.1% reduced sheath blight and blast in rice caused by *R. solani* and *P. grisea* by 92.78% and 100% under detach leaves assay, respectively [59,74].

In the greenhouse trial, CS-NPs are capable of protecting plants of rice, finger millet, and wheat from pathogens attacks [52,71,74]. The sheath blight disease was reduced by 75.01% compared with CS at 44.82%, and the peroxidase, phenylalanine ammonia-lyase, and chitinases activity also increased by 19-, 1.5-, and 1.9-fold, respectively [74]. In the study of [71], the symptom and disease incidence of blast was delayed by 10 days and decreased 2.8-fold, respectively, influenced by peroxidase activity (which increased 1.6-fold) and reactive oxygen species activity. Spray of CS-NPs at 0.05% after infection of *F. graminearum* leads to reduce AUPDC at 28 days after inoculation (DAI) by 2.2-fold compared to the water control. The NPs caused structural damage in mycelium and cell pathogen but also increased superoxide and H_2_O_2_ content [52].

### 3.2. Chitosan-Nanoparticles-Loaded Active Ingredients

#### 3.2.1. Directly

The effect of controlling or enhancing the immunity of plants is different, depending on the same CS-NPs and the type of active ingredient. The EC_50_ of four formulate CS-NPs-loaded hexaconazole to control *Ganoderma boninense* is 8.0–18.4 ppb, which is 21.4 and 1534.5 ppb lower than hexaconazole and CS-NPs, respectively. Similarly, fiducial limit (lower-upper) was 6.0–10.9 to 13.0–32.8 ppb, while hexaconazole and CS-NPs were 16.7–27.3 and 494.0–13280.4 ppb [58].

CS-NPs-loaded Cu could inhibit mycelial growth of *Curvularia lunata* by 50.0 and 52.7% at 0.12 and 0.16% [75], *A. solani* and *F. oxysporum* by 84.2 and 60.1% at 0.1% [76], *A. alternata* and *R. solani* by 82.1–89.5% and 62.5–63.0% at 0.06–0.1%, and *M. phaseolina* by 60.1% at 0.1% [51], respectively. These NPs also inhibited spore germination of *A. solani* and *F. oxysporum* by 73.3 and 79.9% at 0.1% [75] and *A. alternata* by 83.3–87.4% at 0.06–0.1% [51]. CS-NPs-loaded Zn inhibited mycelial growth and spore germination of *C. lunata* by 47.7–65.2% and 50.5–73.3% at 0.08–0.16% [77]. In addition, the mixture of CS-NPs (ionic gelation) and Cu-NPs (chemical reduction) inhibited the mycelial growth of *F. oxysporum* by 61.94–100% at 0.05–0.2% [57]. CS-NPs-loaded SA evaded mycelial growth by 62.2–100% and spore germination of *Fusarium verticillioides* by 48.3–60.5% at 0.08–0.16% [65]. CS-NPs-loaded saponin inhibited mycelial growth of *A. alternata* by 78.3–80.9% and *R. solani* by 27.7% at 0.06–0.1% and spore germination of *A. alternata* by 78.3–82.9% at 0.1% [51]. On the other hand, CS-NPs-loaded thiamine did not inhibit *F. oxysporum*, even at a concentration of 0.1% [66].

Under greenhouse conditions, at 3 DAI, *A. solani* and *F. oxysporum*, CS-NPs-loaded Cu (0.1 and 0.12%) was foliar sprayed and applied to soil lead to reduced early blight (84.2 and 87.7%) and fusarium wilt (49.9 and 61.1%), respectively [75]. Furthermore, priming maize seeds into these NPs (0.02–0.14%) for 4 and 8 h combined with foliar spraying after *F. verticillioides* infected reduced post-flowering stalk rot disease by 38.2–48.1% and 24.8–49.6%, respectively [78]. Moreover, these treatments reduced disease severity by 23.5–33.9% and 2.55–15.8% for 4 and 8 h priming under field conditions.

#### 3.2.2. Indirectly

Previously, Harpin protein (from *Erwinia amylovora*) was known for its ability to induce systemic acquired resistance in plants [79]. With the same amount (20 µg), CS-NPs-loaded Harpin protein (from *P. syringae* pv. *syringae*) enhanced cell death, necrotic lesions, and H_2_O_2_ accumulation faster and stronger than Harpin protein only [60]. Furthermore, treatment of these NPs reduced fungal biomass (5-fold) and lesion diameter (12-fold) and caused failing colonization of *R. solani* in tomato leaves compared with the control. Peroxidase and phenylalanine ammonia-lyase activity also steadily increased up to 72 h. Interestingly, the transcriptome changes, including defense response, signal transduction, transport, transcription, photosynthesis, housekeeping, and aromatic biosynthesis, were enhanced more than 2-fold at 24, 48, and 72 h after spraying.

Under greenhouse conditions, pre-treated CS-NPs-loaded Cu (0.04–0.16%) reduced leaf spot disease (*C. lunata*) in maize by 43.86–48.48%. Moreover, this treatment increased superoxide dismutases (1.8–2.2 folds), peroxidase (1.5–2.1 folds), phenylalanine ammonia-lyase (1.3–2.0 folds), and polyphenol oxidase (1.1–1.2 folds) [75]. Furthermore, CS-NPs-loaded Zn also induced superoxide dismutases, phenylalanine ammonia-lyase, polyphenol oxidase, and H_2_O_2_ activity by 1.2–2.0-, 2.0–3.0-, 17.24–49.37-, and 1.5–2.6-fold when compared with the control, respectively. The H_2_O_2_ and lignin localization also increased, leading to maize leaf spot (*C. lunata*) reduction by 32.3–50.77% [77].

A hormone-elicitor is an SA that has been loaded into CS-NPs. Maize was pre-treated with these NPs at 0.01–0.16% and suppressed post-flowering stalk rot disease (37.33–49.5%) caused by *F. verticillioides*. Furthermore, at 2 and 3 days after spraying NPs, superoxide dismutases (1.8- and 3.2-fold), peroxidase (7.0- and 4.6-fold), catalase (3.1- and 2.6-fold), phenylalanine ammonia-lyase (2.0- and 1.7-fold), polyphenol oxidase (1.7- and 2.0-fold), O_2_^−^ (1.1- and 1.1-fold), H_2_O_2_ (17.5- and 37.0-fold), and lignin accumulation increased [65]. Pre-treated CS-NP-loaded SA at 400 ppm and CS-NP-loaded Ag at 200, 400, and 800 ppm by stalk-soaking and foliar spraying reduced cassava leaf spot disease by 68.9–73.6% at 56 DAP (first inoculate with fungal density 10^4^ conidia per mL) and 37.0–37.7% at 75 DAP (second inoculate with fungal density 10^5^ conidia per mL) [69].

Although CS-NPs-loaded thiamine did not inhibit fungi *in vitro* condition, pre-treatment of these NPs (0.1%) at 3 days before infection of *F. oxysporum* on chickpea reduced cell death in 2 DAI compared with control. Furthermore, polyphenol oxidase, peroxidase, β-1,3-glucanase, chitinase, chitosanase, and protease were increased by 2.1-, 2.0-, 1.4-, 1.4-, 1.4-, and 1.1-fold in leaves and 2.0-, 1.3-, 1.1-, 1.3-, 1.3-, and 1.1-fold in roots, respectively [66].

On the other hand, in the study of [80], CS-NPs-loaded Cu at 0.12–0.06% was treated before and after an infection of *Xanthomonas axonopodis* pv. *glycine* could reduce bacterial pustule disease in soybean by 40.6–49.7%, respectively. Interestingly, the low concentration is more effective. In addition, application of the mixture of CS-NPs (ionic gelation) and Cu-NPs (chemical reduction) to date palm root zone increased plant immunomodulatory, including total phenols (1.1–1.5 folds), phenoloxidases (1.1–2.0 folds), and peroxidase (1.6–3.0 folds), which led to a reduction in disease by 16.2–59.3% [57].

Under field conditions, CS-NPs-loaded Cu, Zn, and SA are effective in reducing disease by inducing plant defense system in maize and soybean [65,75,77,80]. Treatment CS-NPs-loaded Cu (0.06%) reduced bacterial pustule disease by 51.3%. In addition, these NPs at 0.01–0.08% reduced maize leaf spot disease by 27.72–28.53% while at 0.12–0.16% they reduced it by 30.42–33.8% [75]. On the other hand, CS-NPs-loaded Zn at 0.01–0.16% reduced this disease by 25.42–39.67% [77]. CS-NPs-loaded SA at 0.01–0.16% reduced post-flowering stalk rot by 40.5–59.47% [65].

### 3.3. Plant Growth Promotion

A concern for any agrochemical is the safety of plants, environment, farmers, and consumers. In recent reviews, NP is a biosafety solution. However, nanotoxicology still remains to be noticed [81,82]. When applying NPs to plants, they will enter the tissues and cause positive and negative impacts depending on their size, shape, and concentration. NPs usually enhance shoot elongation, root elongation, seed germination at low concentration, and in contrast at high concentration [17]. The effective concentration varies between NPs and crops. In the study by the authors of [69], the CS-NP-loaded SA and silver were tested for phytotoxicity with the cassava by leaf disk assay method before being applied to cassava plants at net house condition. Results showed that these formulations did not cause damage in leaf disk up to 800 ppm. Then, researchers varied concentrations of 25–800 ppm for stalk-soaking and foliar spraying to enhance cassava growth and reduce leaf spot disease. This is an easy way to know what “safe” concentrations are for the plant. When applied to soil, NPs can cause negative impacts on soil microflora but will be less damaging than agrochemical applications [83]. On the other hand, the amount of agrochemical and fertilizer applied to agriculture is reduced if they are replaced by NPs, which leads to a reduction in their toxicity. Usually, the safety-by-design principle is applied to screen potential risks from materials and methods synthesis to NP formulation [84]. As mentioned above, ionic gelation method and CS—a natural polymer—are friendly, safe, and biodegradable solutions.

In addition to its ability to directly inhibit pathogens or induce plant defense system against diseases, CS-NPs or CS-NPs-loaded active ingredients have the ability to stimulate plant growth. At this time, they act as fertilizers or nutrients, affecting plant physiological processes, including nutrient uptake, cell division, cell elongation, enzymatic activation, and the synthesis of protein that leads to increase yield [43]. Efficiency depends on both the CS-NPs and the active ingredient, even when it releases all the active ingredients because the main component of CS is nitrogen, which takes 9–10% [46]. Furthermore, the rich positive charge of CS leads to increased affinity toward the plant cell membrane, which enhances reactivity in the plant system [49].

Several types of NPs presented in Table 3 have been shown to stimulate plant growth.

In the seeding stage, CS-NPs increased the seeding vigor index (57.1%), the number of lateral root (133.3%), and dry weight (200%) of chickpeas [63]. Additionally, the chickpea seeds were soaked with CS-NPs-loaded thiamine at 0.1% overnight, leading to the seeding vigor index increasing by 64.2%, with Indole-3-acetic acid content increased 10-fold [66]. Treatment with CS-NPs-loaded Cu at 0.08, 0.1, and 0.12% increased seedling vigor index (33.9, 33.7, and 24.3%), fresh weight (18.9, 21.6, and 16.2%), and dry weight (20.0, 26.7, and 13.3%) in tomato, respectively [75]. Additionally, CS-NPs-loaded Cu at 0.01–0.16% increased seeding vigor index (15.6–48.6%), fresh weight (7.1–11.4%), and dry weight (21.4–57.1%) in maize seedings, which were related to increasing α-amylase and proteases at days 5 and 7, respectively [85].

Under greenhouse conditions, the dry weight and yield of finger millet increased by 148.8% and 93.2% when treated with CS-NPs, respectively [71]. The plant height, stem height, and root length of maize increased by 30.3–60.3, 66.3–237.5, and 2.7–61.1% when treated with CS-NPs-loaded Zn at 0.01–0.16%, respectively [77]. In chickpea, the shoot length, number of leaves per plant, fresh weight, dry weight, and number of secondary roots per plant increased by 15.3, 14.4, 37.7, 20.0, and 52.8% when sprayed with CS-NPs-loaded thiamine at 0.1%, respectively [66]. In present year, pre-treated CS-NP-loaded SA at 400 ppm and CS-NP-loaded Ag at 200, 400, 800 ppm by stalk-soaking for 5 min and foliar spraying at 28, 42 DAP could increase the number of leaves (45.1–82.4%), the number of shoots (38.5–46.2%), the largest leaf area (29.6–41.9%), root length (11.6–29.9%), and root weight (27.6–82.8%) of cassava, at 75 DAP in net house condition [69]. In addition, CS-NPs-loaded Cu at 0.06% could increase plant height (56.8%), root length (40.3%), and pod number (7.2%). NPs treatment at 0.02% could increase root weight (46.8%), nodule number (44.2%), nodule weight (125.8%) under greenhouse conditions and also increase root length (60.9%), root weight (46.8%), and pod number (29.7%), in soybean under field conditions [80]. Furthermore, NPs at 0.01–0.08% increased plant height (15.9–47.0%), stem diameter (82.9–102.9%), root length (9.5–15.8%), root number (20.9–46.3%), and chlorophyll content (67.3–182.6%) under greenhouse conditions and increased grain yield (25.4–29.3%), 100 grain weight (14.4–16.9%) in maize under field conditions [75]. However, the treatment at 0.16% reduced root length (9.8%) and chlorophyll content (4.6–9.7%), although the difference was not significance. The CS-NPs-loaded SA at 0.01–0.16% increased leaf area (160.6–224.7%), shoot length (38.5–76.9%), root length (66.9–111.5%), root number (59.6–91.8%), stem diameter (22.8–53.9%), and total chlorophyll (54.2–141.4%) in maize under greenhouse conditions. Moreover, the treatment of these NPs at 0.08%, days to 50% tasselling was early by 4 days under field conditions. Moreover, the plant height (25.5%), ear height (12.1%), cob length (44.8%), test weight (71.1%), and grain yield (48.3%) also increased [65]. The authors of [86] synthesized CS-NPs-loaded NPK (ionic gelation) with slow-release N (66.7%), P (3.1%), and K (57.7%) for 240 h. The leaf number, leaf area, plant height, and stem diameter of coffee increased 22.8, 46.9, 12.7, and 28.3% when these NPs were treated at 30 ppm, respectively. This synthesized CS-NPs-loaded NPK improved N (17.04%), P (13.1%), K (67.5%), chlorophyll (30.68%), carotenoid (21.4%) content, and photosynthesis rate (71.7%) in the coffee leaves. Another nano fertilizer, CS-NPs-loaded silicon at 0.04–0.12%, increased the seeding vigor index in maize seeding by 167.5–285.2%. Furthermore, foliar spraying induced antioxidant defense enzyme activity; equilibrated cellular redox; and balanced O_2_^−^ and H_2_O_2_ in leaf, leading to homeostasis. In the field trial, the yield and test weight of maize was increased by 186.6 and 77.1% by treated CS-NPs-loaded silicon at 0.08 and 0.04%, respectively [87].

**Figure 4 polymers-14-00662-f004:**
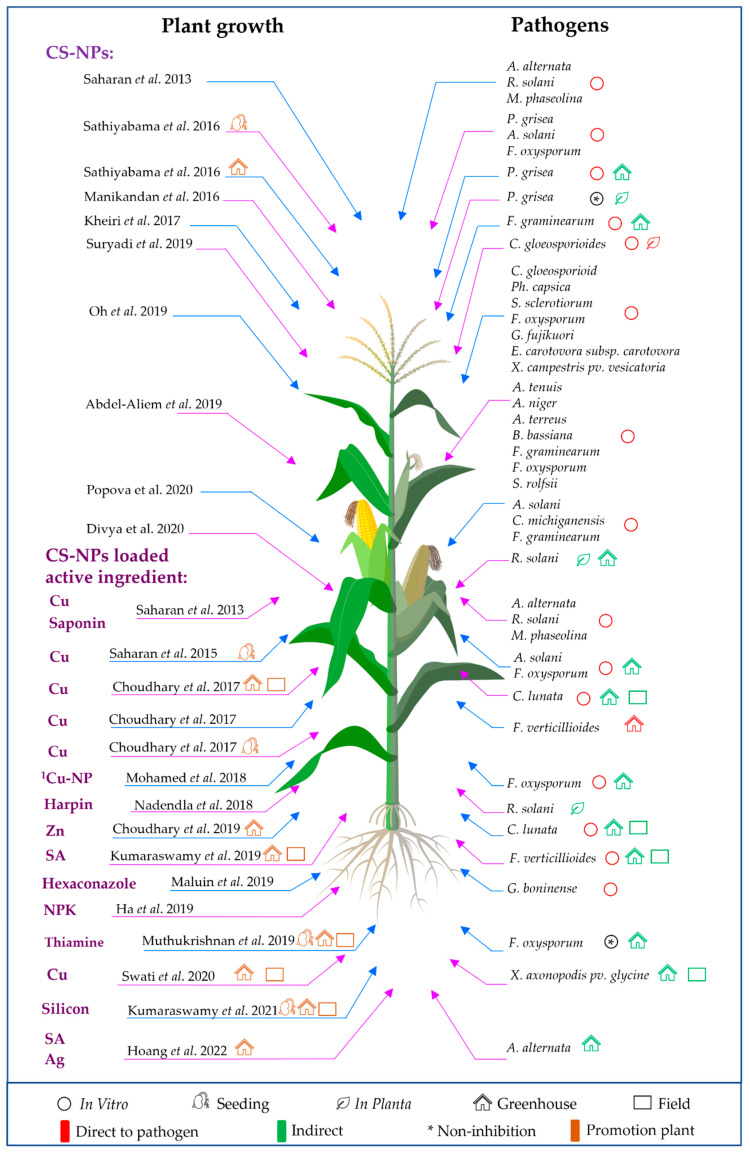
The application of CS-NPs and CS-NPs-loaded active ingredients synthesized by ionic gelation method in plant disease management and enhancing plant growth. Note: ^1^ Mixture of CS-NP (ionic gelation method) and Cu-NP (chemical reduction method). Saharan et al. 2013 [51], Sathiyabama et al. 2016 [63], Sathiyabama et al. 2016 [71], Manikandan et al. 2016 [59], Kheiri et al. 2017 [52], Suryadi et al. 2019 [56], Oh et al. 2019 [54], Abdel-Aliem et al. 2019 [73], Popova et al. 2020 [72], Divya et al. 2020 [74], Saharan et al. 2015 [76], Choudhary et al. 2017 [75], Choudhary et al. 2017 [78], Choudhary et al. 2017 [85], Mohamed et al. 2018 [57], Nadendla et al. 2018 [60], Choudhary et al. 2019 [77], Kumaraswamy et al. 2019 [65], Maluin et al. 2019 [58], Ha et al. 2019 [86], Muthukrishnan et al. 2019 [66], Swati et al. 2020 [80], Kumaraswamy et al. 2021 [87], Hoang et al. 2022 [69].

**Table 3 polymers-14-00662-t003:** The CS-NPs synthesized by ionic gelation used in plant disease management.

NPs	Plant	Pathogen	Summary Research	Reference
CS-NPs with DLS (83.32 nm, PDI 0.31, −28 mV), HRTEM (20–50 nm).	Rice	*P. grisea* /Blast	**In vitro:** Treat CS-NPs not cause inhibit mycelial and spore germination even 0.1%. **Detach leaves assay:** painting brush 500 µL onto surface each leave. After 24 h, inoculate with similar method. Treating CS-NPs 0.1% could prevent blast symptoms up to 10 DAI (suppression 100%).	[59]
CS-NPs described as [59]	Fingermillet	*P. grisea* /Blast	**In vitro:** CS-NPs at 0.1% inhibited nearly 65% mycelial growth and did not sporulate compared with the control. **Greenhouse:** seed was soaked overnight, foliar sprayed at 20 and 30 DAP, and inoculated at 30 DAP. Treatment with CS-NPs delay symptom by 10 days and decreased disease incidence 2.8-fold. Moreover, treatment with CS-NPs increased dry weight (148.8%), yield (93.2%), peroxidase (1.6-fold), and reactive oxygen species activity.	[71]
CS-NPs with DLS (9.8 nm, PDI 0.225, −37 mV), HRTEM (10–30 nm).	Chickpea	*P. grisea* *A. solani* *F. oxysporum*	**In vitro:** CS-NPs with amount 100 µg inhibited mycelial growth of *P. grisea*, *A. solani*, *F. oxysporum* by 92, 87, and 72%, respectively. **Seeding:** Treatment with NPs increased seedling vigor index (57.1%), number lateral root (133.3%), and dry weight (200%).	[63]
CS-NPs Centrifuge method with DLS (180.9 nm, PDI 0.31, and 45.6 mV). pH change method with DLS (225.7 nm, PDI 0.44, and 33.4 mV).	Wheat	*F. graminearum* /Head blight	**In vitro:** CS-NPs synthesized by CS low molecular weight significantly inhibited pathogens, more than low molecular weight and high molecular weight at the same concentration. MIC of CS-NPs prepared by centrifuge and pH change method were 0.05% and 0.09% with 31.97% and 29.67%, respectively. **Greenhouse:** Spraying after inoculate The area under the disease progress curve at 28 DAI of treatment CS-NPs 0.05% was reduced 2.2-fold compared with the control. The CS-NPs caused structural damage in mycelium and cell pathogen and increased superoxide and H_2_O_2_ content.	[52]
CS-NPs with DLS (126.2 nm, PDI 0.44, and 27.8 mV).	Chilli Papaya	*C. gloeosporioides* /Anthracnose	**In vitro:** CS-NPs inhibited mycelial growth (85.7%) and spore germination (61.2%). **In vivo:** preventive (soaking onto CS-NPs 2 mL/L for 60 min before inoculate), curative (soaking onto spore suspension for 15 min following air-dried and soaking onto CS-NPs for 60 min). Inhibition rates of preventive and curative treatment were 87.5% and 75% for chilli and 50% and 10% for papaya.	[56]
CS-NPs with size 100 nm	Tomato	*C. gloeosporioides*, *P. capsici*, *S. sclerotiorum*, *F. oxysporum*, *G. fujikuori*, *E. carotovora* subsp. *carotovora*, *X. campestris* pv. *vesicatoria*	**In vitro:** CS-NPs at 0.5% inhibited mycelial growth of *C. gloeosporioides*, *P. capsici*, *S. sclerotiorum*, *F. oxysporum*, and *G. fujikuori* by 37.8, 50.7, 39.5, 50.3, and 56.3% at day 10 (except *S. sclerotiorum* was at day 5), respectively. CS-NPs at 0.5% reduced OD_600 nm_ of *E. carotovora* subsp. *carotovora* strains 113114, 113154, and YKB133061, and *X. campestris* pv. *vesicatoria* strain 11,154, by 41.3, 55.5, 48.5, and 52.1, respectively. Interestingly, CS-NPs at 0.05% were similarly reduced by 64.7, 76.3, 78.0, 73.8%, respectively.	[54]
CS-NPs	Rice	*R. solani* /Sheath blight	**Detach leaves assay:** Pre-treated CS-NPs and CS at 0.1% reduced disease leaf area by 92.78% and 78.89%, respectively. **Greenhouse:** Seed treat 2 h, soil amended and foliar spraying 15 and 30 DAP, inoculate 45 DAP The disease was suppressed 75.01% and 44.82% when treated with CS-NPs and CS, respectively. Peroxidase, phenylalanine ammonia-lyase, and chitinase activity were increased 19-, 1.5-, and 1.9-fold, respectively.	[74]
CS-NPs with DLS (47 nm, PDI 0.45, and 26.8 mV)	Tomato Cereal	*C. michiganensis* /Bacterial canker *A. solani* /Leaf spot *F. graminearum* /Head blight and root rot	**In vitro:** lysis zone diameters of *C. michiganensis-* and *F. graminearum*-treated CS-NPs at 0.014% (in acetate buffer) were 29.5 and 20 mm, respectively. Similarly, CS were 22.5 and 18.0 mm. CS-NPs at 0.03 and 0.04% (in acetate buffer) inhibited mycelial of *A. solani* by 10 and 70% compared with CS, respectively.	[72]
CS-NPs with DLS (192.5 nm, PDI 0.6, +45.33 mv) CS-NPs-loaded saponin with DLS (373.9 nm (2 peaks), PDI 1.0, +31 mV) And CS-NPs-loaded copper (Cu) with DLS (196.4 nm, PDI 0.5, +88 mV)	-	*A. alternata* *M. phaseolina* *R. solani*	**In vitro:** CS-NPs, CS-NPs-loaded saponin, and CS-NPs-loaded copper at 0.06–0.1% inhibited mycelial growth of *A. alternate* (80.1–82.2, 78.3–80.9, and 82.1–89.5%) and *R. solani* (32.2–34.4, 27.7, and 62.5–63.0%) respectively. For *M. phaseolina*, 3 NPs at 0.1% inhibited 84, 66.2, and 60.1%, respectively. Moreover, 3 NPs at 0.06–0.1% inhibited *A. alternate* spore germination by 84.4–87.1, 78.3–82.9, and 83.3–87.4%, respectively.	[51]
CS-NPs-loaded copper (Cu) with DLS (374.3 nm, PDI 0.33, and 22.6 mV), TEM (150 nm)	Tomato	*A. solani* /Early blight *F. oxysporum* /Wilt	**Seeding:** Treatment with CS-NPs-loaded Cu at 0.08, 0.1, and 0.12% increased seedling vigor index (33.9, 33.7, and 24.3%), fresh weight (18.9, 21.6, and 16.2%), and dry weight (20.0, 26.7, and 13.3%), respectively. **In vitro:** CS-NPs-loaded Cu at 0.1% inhibited mycelium growth and spore germination of *A. solani* and *F. oxysporum* by 84.2% and 60.1%, and 73.3% and 79.9%, respectively. **Greenhouse:** Spray at 3–4 DAI (*A. solani*); apply to soil at 3 DAI (*F. oxysporum*). NPs at 0.1 and 0.12% reduced early blight at 84.2% and 87.7%, fusarium wilt at 49.9%, and 61.1%, respectively.	[75]
CS-NPs-loaded copper (Cu) with DLS (295.4 nm, PDI 0.28, and 19.6 mV)	Maize	*F. verticillioides* /Post flowering stalk rot	**Greenhouse:** Inoculate before planting by mix with soil, seed treat for 4, 8 h, and spray at 45 and 65 DAP. NP treatment at 0.02–0.14% reduced disease severity 38.2–48.1% and 24.8–49.6% for seeds treated for 4 and 8 h, respectively. **Field:** similar greenhouse experiments except inoculate at flowering stage. NP treatment at 0.02–0.14% reduced disease severity by 23.5–33.9% and 2.55–15.8% for seed treated for 4 and 8 h, respectively.	[78]
CS-NPs-loaded copper (Cu) with DLS (361.3 nm, PDI 0.2, and 22.1 mV)	Maize	*C. lunata* /Leaf spot	**In vitro:** CS-NPs-loaded Cu at 0.12 and 0.16% inhibited mycelial growth at 50.0 and 52.7%, respectively. **Greenhouse:** Treat seeds for 4 h, foliar spray 35 DAP, and inoculate 45 DAP. Treatment with CS-NPs-loaded Cu at 0.01–0.08% increased plant height (15.9–47.0%), stem diameter (82.9–102.9%), root length (9.5–15.8%), root number (20.9–46.3%), and chlorophyll content (67.3–182.6%). However, treatment at 0.16% reduced root length (9.8%) and chlorophyll content (4.6–9.7%), although the difference was non-significant. Moreover, treatment with NPs at 0.04–0.16 increased superoxide dismutases (1.8–2.2-fold), peroxidase (1.5–2.1-fold), phenylalanine ammonia-lyase (1.3–2.0-fold), and polyphenol oxidase (1.1–1.2-fold, which also reduced disease severity 43.86–48.48%). **Field:** Treatment with CS-NPs-loaded Cu at 0.01–0.08% reduced disease severity by 27.72–28.53%. Similarly, NPs at 0.12–0.16% reduced it by 30.42–33.88% and increased grain yield (25.4–29.3%), 100 grain weight (14.4–16.9%).	[75]
CS-NPs with DLS (86.8 nm, 32.4 mV), CS-NPs load Harpin (*P. syringae* pv. *syringae*), and DLS (133.7 nm, 48.6 mV)	Tomato	*R. solani*	**In planta:** Treated amount of 20 µg of CS-NPs load Harpin enhanced cell death, necrotic lesion, and H_2_O_2_ accumulation faster and stronger than Harpin protein only. Moreover, treatment with NPs reduced fungal biomass (5 folds), lesion diameter (12 folds), and failed colonization in leaves, when compared with control. For mechanism, peroxidase and phenylalanine ammonia-lyase activity steadily increased up to 72 h. The transcriptome change, including defense response, signal transduction, transport, transcription, photosynthesis, housekeeping, and aromatics biosynthesis, was enhanced more than 2-fold at 24, 48, and 72 h after spraying.	[60]
CS-NPs 50 nm	Date palm	*F. oxysporum* /Vascular wilt	Mix CS-NPs (ionic gelation method) and Cu-NPs (chemical reduction method) to obtain copper-chitosan nanocomposition (CuCs) **In vitro:** CuCs at 0.05–0.2% could inhibit 61.94–100% mycelial growth. **Greenhouse:** Apply 50 mL of CuCs to root zone of seeding. Treated CuCs increased plant immunomodulatory, including total phenol (1.1–1.5 folds), phenoloxidase (1.1–2.0 folds), and peroxidase (1.6–3.0 folds), which led to reduced disease by 16.2–59.3%.	[57]
CS-NPs (DLS 180 nm with range 500–800 nm)	Ground nut oil seed	*A. tenuis* *A. niger* *A. terreus* *B. bassiana* *F. graminearum* *F. oxysporum* *S. rolfsii*	**In vitro:** CS-NP at 800 ppm inhibited mycelial growth of *A. tenuis, A. niger, A. terreus, B. bassiana, F. graminearum, F. oxysporum,* and *S. rolfsii* by 67.67, 62.75, 74.67, 76.08, 60.37, 66.60, and 37.41%, respectively. Moreover, 0.2 mL of CS-NP at 125 ppm inhibited *F. graminearum* by 44.3%, higher than fungicide (8-hydroxy quinoline), which was 42.33%. In addition, the CS-NP at 800 ppm reduced zearalenone secreted by *F. graminearum.*	[73]
CS-NPs-loaded zinc (Zn) with DLS (387 nm, PDI 0.22, and 34 mV), TEM/SEM (200–300 nm, spherical)	Maize	*C. lunata*	**In vitro:** NPs could inhibit mycelium growth at 47.7–65.2% and 0.08–0.16% and spore germination at 50.5–73.3% and 0.01–0.16%. **Greenhouse:** Seed treat 4 h, foliar spraying 35 DAP, inoculate 45 DAP. The superoxide dismutases, phenylalanine ammonia-lyase, polyphenol oxidase, H_2_O_2_ activity could be increased at 1.2–2.0, 2.0–3.0, 17.24–49.37, 1.5–2.6 folds when compared with the control, respectively. H_2_O_2_ and lignin localization were also increased. The DS was reduced 32.3–50.77%. The plant height, stem diameter, root length were increased 30.3–60.3, 66.3–237.5, 2.7–61.1%, respectively. **Field:** The DS was reduced at 25.42–39.67%.	[77]
CS-NPs-loaded SA with DLS (368.7 nm, PDI 0.1, and 34.1 mV)	Maize	*F. verticillioides* /Post-flowering stalk rot	**In vitro:** CS-NPs-loaded SA treatment at 0.08–0.16% could evade mycelial growth at 62.2–100% and spore germination at 48.3–60.5%. **Greenhouse:** Treat seeds for 4 h, foliar spray 55 DAP, and inoculate 60 DAP. NP treatment at 0.01–0.16% reduced disease severity at 37.33–49.5% and increased leaf area (160.6–224.7%), shoot length (38.5–76.9%), root length (66.9–111.5%), root length (59.6–91.8%), stem diameter (22.8–53.9%), and total chlorophyll (54.2–141.4%). Moreover, at 2 and 3 days after spraying NPs, superoxide dismutases (1.8- and 3.2-fold), peroxidase (7.0- and 4.6-fold), catalase (3.1- and 2.6-fold), phenylalanine ammonia-lyase (2.0- and 1.7-fold), polyphenol oxidase (1.7- and 2.0-fold), O_2_^−^ (1.1- and 1.1-fold), H_2_O_2_ (17.5- and 37.0-fold), and lignin accumulation also increased. **Field:** NP treatment reduced disease severity at 40.5–59.47%. At 0.08%, 50% tasseling was early by 4 days. Moreover, the plant height (25.5%), ear height (12.1%), cob length (44.8%), test weight (71.1%) and grain yield (48.3%) also increased.	[65]
CS-NPs with DLS (bimodal particle with 2.3 and 7.5 nm), TEM (1.5 nm), and CS-NPs-loaded [H]hexaconazole, as described in Table 1	Oil palm	*G. boninense*	**In vitro:** EC_50_ of hexaconazole, CS-NPs, and four formulate CS-NPs-loaded hexaconazole were 21.4, 1534.5, 8.0, and 18.4 ppb, respectively. Similar, fiducial limit (lower-upper) was 16.7–27.3, 494.0–13280.4, 6.0–10.9 and 13.0–32.8 ppb, respectively.	[58]
CS-NPs-loaded Thiamine with DLS (596 nm, 37.7 mV), HRTEM (10–60 nm)	Chickpea	*F. oxysporum* /Wilt	**Seeding:** soaking seed overnight at 0.1% increased seedling vigor index by 64.2% and Indole-3-acetic acid content 10-fold. **In vitro**: CS-NPs-loaded thiamine did not inhibit fungi even by 0.1%. **Greenhouse:** Spraying 12 DAP and inoculating 15 DAP NP treatment at 0.1% reduced cell death in 2 DAI compared with the control. Moreover, shoot length, number of leaves per plant, fresh weight, dry weight, and number of secondary roots per plant also were increased by 15.3, 14.4, 37.7, 20.0, and 52.8%, respectively. In leaves, polyphenol oxidase, peroxidase, β-1,3-glucanase, chitinase, chitosanase, and protease were increased by 2.1-, 2.0-, 1.4-, 1.4-, 1.4-, and 1.1-fold, respectively. In root, this enzyme activity was increased 2.0-, 1.3-, 1.1-, 1.3-, 1.3-, and 1.1-fold, respectively.	[66]
CS-NPs-loaded copper (Cu) with DLS (314 nm, PDI 0.48, and 19.5 mV)	Soybean	*X. axonopodis* pv. *glycine* /Bacterial pustule	**Greenhouse:** Seed treatment for 4 h; foliar spraying at trifoliate stage; and, after disease occurrence, inoculate 35 DAP. CS-NPs-loaded Cu at 0.12–0.06% reduced disease by 40.6–49.7%. NP treatment at 0.06% increased plant height (56.8%), root length (40.3%), and pod number (7.2%). NP treatment at 0.02% increased root weight (46.8%), nodule number (44.2%), and nodule weight (125.8%). **Field:** CS-NPs load Cu treatment at 0.06% reduced disease by 51.3% and increased root length (60.9%), root weight (46.8%), and pod number (29.7%).	[80]
CS-NP-loaded SA with DLS (89.86 nm, PDI 0.36, and 22.27 mV), CS-NP-loaded silver (Ag) with DLS (249 nm, PDI 0.53, and 13.53 mV)	Cassava	*A. alternata* /Leaf spot	**Leaf disk assay:** two NP formulations not caused phytotoxicity upto 800 ppm. **Greenhouse:** Cassava stalk-soaking for 5 min, foliar spraying at 28 and 42 DAP, and inoculate 44 DAP with density 10^4^ conidial per mL and 63 DAP with density 10^5^ conidial per mL. CS-NP-loaded SA at 400 ppm and CS-NP-loaded Ag at 200, 400, and 800 ppm reduced disease by 68.9–73.6% at 56 DAP (first inoculate) and by 37.0–37.7% at 75 DAP (second inoculate). These treatments increased the number of leaves (45.1–82.4%), the number of shoots (38.5–46.2%), the largest leaf area (29.6–41.9%), root length (11.6–29.9%), and root weight (27.6–82.8%).	[69]

Note: CS: chitosan; DAI: days after inoculate; DAP: days after planting; HRTEM: high-resolution transmission electron microscopy; NPs: nanoparticles; PDI: polydispersity index; SEM: scanning electron microscope; TEM: transmission electron microscopy.

## 4. Conclusions and Future Perspectives

Discovered since 1997, the studies on NPs synthesized by ionic gelation method have only received attention in the last ten years. The researches on using these NPs in plant disease management has only been interested in the last five years. With the advantage of being easy to implement, both CS-NP and CS-NP-loaded active ingredients (Cu, Saponin, Harpin protein, Zn, SA, Hexaconazole, NPK, Thiamine, Silicon, and Ag) are effective in plant disease management and enhancing plant growth depending on the concentration and application method by direct or indirect mechanisms. CS-NP-loaded active ingredients constitute the “drug delivery system” model. The effectiveness of disease management and enhanced plant growth of CS-NP or CS-NP depend on the mechanism of CS (carrier) and active ingredients (drug). At higher concentrations, CS-NP or CS-NP-loaded active ingredients are effective in directly inhibiting phytopathogens. This can be applied to control when the disease has broken out. In addition, CS-NP and CS-NP-loaded active ingredients at lower concentrations can indirectly reduce disease through activation of plant’s innate immunity, including stimulating cell death, H_2_O_2_ accumulation, oxidative burst (O_2_^−^), enzymes (β-1,3-glucanase, catalase, chitinase, chitosanase, peroxidase, phenoloxidases, phenoloxidases, phenylalanine ammonia-lyase, polyphenol oxidase, protease, and superoxide dismutases), and secondary metabolites (total phenols, lignin). Moreover, their treatment can enhance transcriptome changes, including defense response, signal transduction, transport, transcription, photosynthesis, housekeeping, and aromatic biosynthesis. In nature, plant diseases often have seasonal outbreaks. Periodical pre-treat CS-NP or CS-NP-loaded active ingredients at sensitive periods can prevent disease and reduce the consequences of disease outbreaks. Furthermore, CS-NP and CS-NP-loaded active ingredients can enhance indole-3-acetic acid, α-amylase, protease, chlorophyll, carotenoid content, and photosynthesis rates, leading to increased plant growth, yield, and quality. When plants grow well, their health is enhanced and they can better tolerance diseases. In particular, CS-NP and CS-NP-loaded active ingredients are nano-sized, have a positive charge, and are able to easily penetrate cells or stick to plant surfaces. Moreover, the active ingredient can be slowly released into plant and easily absorbed with no waste. The CS (carrier) as a nitrogen source enhances cell division, cell elongation, enzymatic activation, and synthesis of protein. These preeminent characteristics lead to CS-NP or CS-NP-loaded active ingredients being more effective than CS or active ingredients alone. The CS-NP-loaded active ingredients are more interested in evaluating effectiveness in greenhouse and filed conditions. Most of the studies are more interested in fungal diseases (*Alternaria* spp., *Aspergillus* spp., *B. bassiana, C. gloeosporioides, C. lunata, Fusarium* spp., *G. boninense, G. fujikuori, M. phaseolina, P. capsici, P. grisea, R. solani, S. sclerotiorum,* and *S. rolfsii*) and bacteria (*C. michiganensis, E. carotovora* subsp. *carotovora,* and *Xanthomonas* spp.) than viruses, phytoplasma, viroid, and nematode. Many crops, including cassava, chickpea, chilli, date palm, fingermillet, maize, papaya, rice, soybean, tomato, and wheat, have been evaluated in in vivo or greenhouse conditions. However, field experiments are still limited as only maize (with CS-NP-loaded Cu, Zn, SA, and Silicon) and soybean (with CS-NP-loaded Cu) have been evaluated for managing post-flowering stalk rot, Curvularia leaf spot, and bacterial pustule disease and/or enhancing plant growth.

Nanotechnology is the trend of the future. Easy access and dissemination of nano pesticides are essential, especially in developing areas. Since 2019, five of eight studies performed in field conditions have shown interest in CS-NPs synthesized by the ionic gelation method. In the future, new active ingredients could be loaded into CS-NPs or new polymers with anions by ionic gelation methods and used to improve crop yields. A hypothesis is proposed that “mixing CS and TPP under stirring conditions will lead to CS-NPs formation”; then, character or not, they will still be NPs and possess the superiority of NPs. Therefore, the “legendary” pairs of counter ions, CS and TPP, can be studied for immediate application in fields in developing regions where advanced research facilities are limited to building sustainable agriculture.

## Figures and Tables

**Figure 1 polymers-14-00662-f001:**
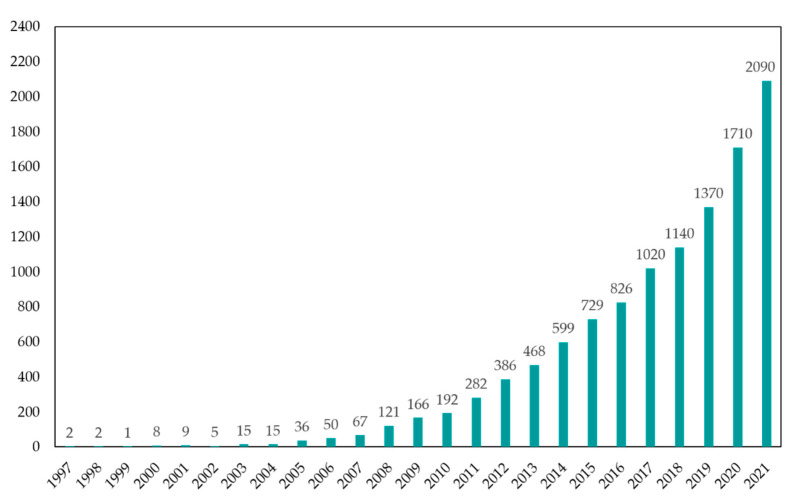
The number of articles in Google Scholar searched by key word “ionic gelation” + “nanoparticles” [39].

**Figure 2 polymers-14-00662-f002:**
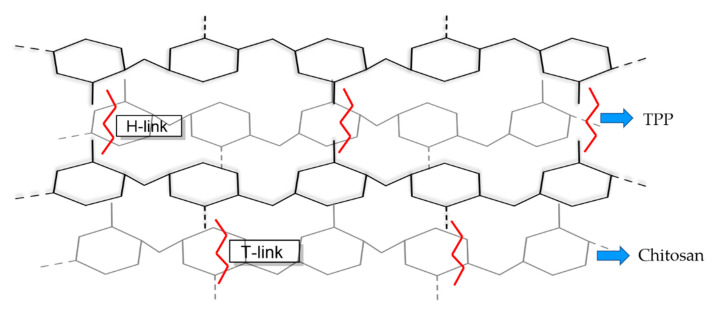
The electrostatic interactions between CS and TPP by an H-link configuration (in the same plane) and T-link configuration (in a different plane) in ionotropic gelation.

**Figure 3 polymers-14-00662-f003:**
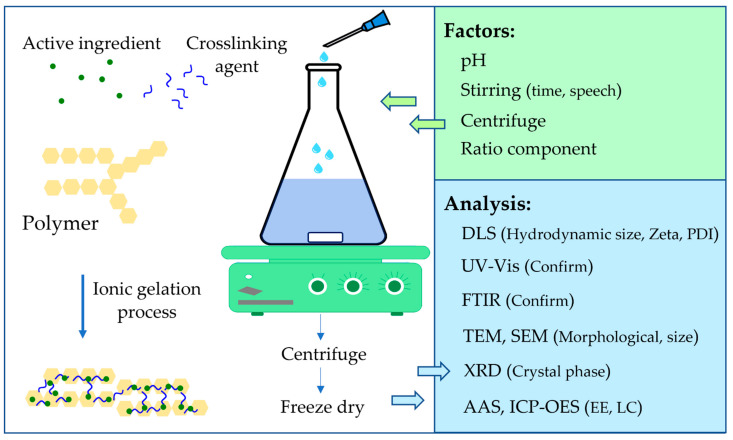
The schematic representation of nanoparticles synthesized and characterized by ionic gelation method.

## Data Availability

Not applicable.

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
