# Peer review of "Chitosan Nanoparticles-Based Ionic Gelation Method: A Promising Candidate for Plant Disease Management"

_polymers, 2022, doi:10.3390/polym14040662_

Round 1

Reviewer 1 Report

The authors well presented their information by proper images. Chitosan nanoparticles prepared by the ionic gelation method were introduced as a promising candidate for plant disease management. I suggest acceptance for this review after a minor revision

  1. in Tables 1 and 2, please add a hyphen (-) for those are empty
  2. please discuss cytotoxicity issues of NPs in plant
  3. introduction need more discussion of newly published papers for example (10.1007/s10725-021-00782-w, 10.1016/j.ijbiomac.2021.11.028, 10.2147/IJN.S318416 and 10.3390/molecules26061770

Reviewer 2 Report

The submitted manusript is the review dealing with the chitosan based nano-particles, methods of their preparation and possible use for plant disease management. Sufficient number of relevant references is cited. The review seems to be the detailed summary of works studied the ionic gelation as the method of preparation of nano-particles. The mechanism of cross-linking is described. 

I would like to ask authors for detail desription of mechanism of effect of chitosan nanoparticles on plant disease managemment and plant growth.

In my oponion, the "Conclusion and Future perspectives" should be more concrete and aimed to real utilization of gather knowledge.
